# Aqueous pK$_a$ prediction for tautomerizable compounds using equilibrium bond lengths

Beth A. Caine[1,2], Maddalena Bronzato[3], Torquil Fraser[3], Nathan Kidley[3], Christophe Dardonville [4] & Paul L.A. Popelier [1,2 ✉]

The accurate prediction of aqueous pK$_a$ values for tautomerizable compounds is a formidable task, even for the most established in silico tools. Empirical approaches often fall short due to a lack of pre-existing knowledge of dominant tautomeric forms. In a rigorous first-principles approach, calculations for low-energy tautomers must be performed in protonated and deprotonated forms, often both in gas and solvent phases, thus representing a significant computational task. Here we report an alternative approach, predicting pK$_a$ values for herbicide/therapeutic derivatives of 1,3-cyclohexanedione and 1,3-cyclopentanedione to within just 0.24 units. A model, using a single ab initio bond length from one protonation state, is as accurate as other more complex regression approaches using more input features, and outperforms the program Marvin. Our approach can be used for other tautomerizable species, to predict trends across congeneric series and to correct experimental pK$_a$ values.

[1] Department of Chemistry, University of Manchester, Manchester, UK. [2] Manchester Institute of Biotechnology (MIB), 131 Princess Street, Manchester, UK. [3] Syngenta AG, Jealott's Hill, Warfield, Bracknell RG42 6E7, UK. [4] Instituto de Química Médica, IQM–CSIC, C/Juan de la Cierva 3, Madrid 28006, Spain. ✉email: paul.popelier@manchester.ac.uk

Approximately 21% of the compounds that make up pharmaceutical databases are said to exist in two or more tautomeric forms[1]. Tautomerism is a form of structural isomerism that is characterized by a species having two or more structural representations, between which interconversion can be achieved by "proton hopping" from one atom to another. Issues surrounding $pK_a$ prediction for species exhibiting this feature have been noted a number of times in the literature. Recently[2], Connolly suggested that a lack of experimental information on both relative tautomer stability and the properties of distinct tautomeric forms are the likely causes of such issues. Tautomeric species present a challenge, not just to empirical-based approaches, but also to those that attempt to solve the $pK_a$ prediction problem using first-principles[1–5]. For tools implementing the latter approach (e.g. Jaguar, Schrödinger[4,6,7]), the most rigorous protocol includes quantum chemical calculations for conformations of each, or a select few low lying tautomer(s), in both gas- and solvent phase, and in both protonated and deprotonated forms. Therefore, without some element of empiricism, first-principles approaches often incur significant computational expense.

For methods of $pK_a$ estimation that generate descriptors starting from 2D fingerprints, each tautomeric form of a species will correspond to a unique representation. Therefore, the user must either (i) possess prior knowledge of tautomeric stability in order to maximize prediction accuracy, or (ii) tautomer enumeration must be performed by the program based on an arbitrary user input, followed by selection of the optimal tautomer for calculation of chemical descriptors[8–10]. A comparative study[11] of 5 empirical $pK_a$ prediction tools (ACD/$pK_a$ DB (http://www.acdlabs.com/home), Epik (http://www.schrodinger.com), VCC (http://vcclab.org), Marvin (http://www.chemaxon.com) and Pallas (www.compudrug.com)) on 248 compounds of the Gold Standard Dataset compiled by Avdeef[12], demonstrated a tendancy for prediction errors to be higher for compounds with a larger number of possible tautomeric states. For the tool they tested, the guanidine group of the drug Amiloride and the enolic hydroxyl groups of herbicides Sethoxydim and Tralkoxydim were also identified as common outliers.

Compounds containing a 1,3-diketo group exhibit tautomerism (shown in Fig. 1a(i), (ii)). For cyclic 1,3-diketones, the diketo state (Fig. 1a(i)) can be transformed into two keto-enol forms (Fig. 1a(ii)). Tautomeric states of the same molecule may be non-degenerate, with the ratio being influenced by the solvent environment and temperature[13]. The compounds 1,3-cyclohexanedione (1,3-CHD) and 1,3-cyclopentanedione (1,3-CPD) are known to possess significant keto-enol character in solution, a phenomenon attributed to the formation of hydrogen bonded solute dimers, and additional stabilization from solute-solvent interactions[14].

1,3-CHD is a fragment prevalent to both agrochemically and pharmaceutically active compounds in use today. Alloxydim (Fig. 1b(i)) is currently used as a selective systemic herbicide for post-emergence control of grass weeds in sugar beet, vegetables and broad-leaved crops. Adding a derivatized benzoyl group at the 2-position in place of Alloxydim's 2-oxime forms what is known as triketone herbicide (e.g. Mesotrione, Fig. 1b(ii)). Pharmaceutically relevant compounds containing the 1,3-CHD group include the antibiotic Tetracycline and its analogues.

Previous work[15–22] from our group, as well as the earlier work of others, has highlighted the utility of bond lengths[23–25] and

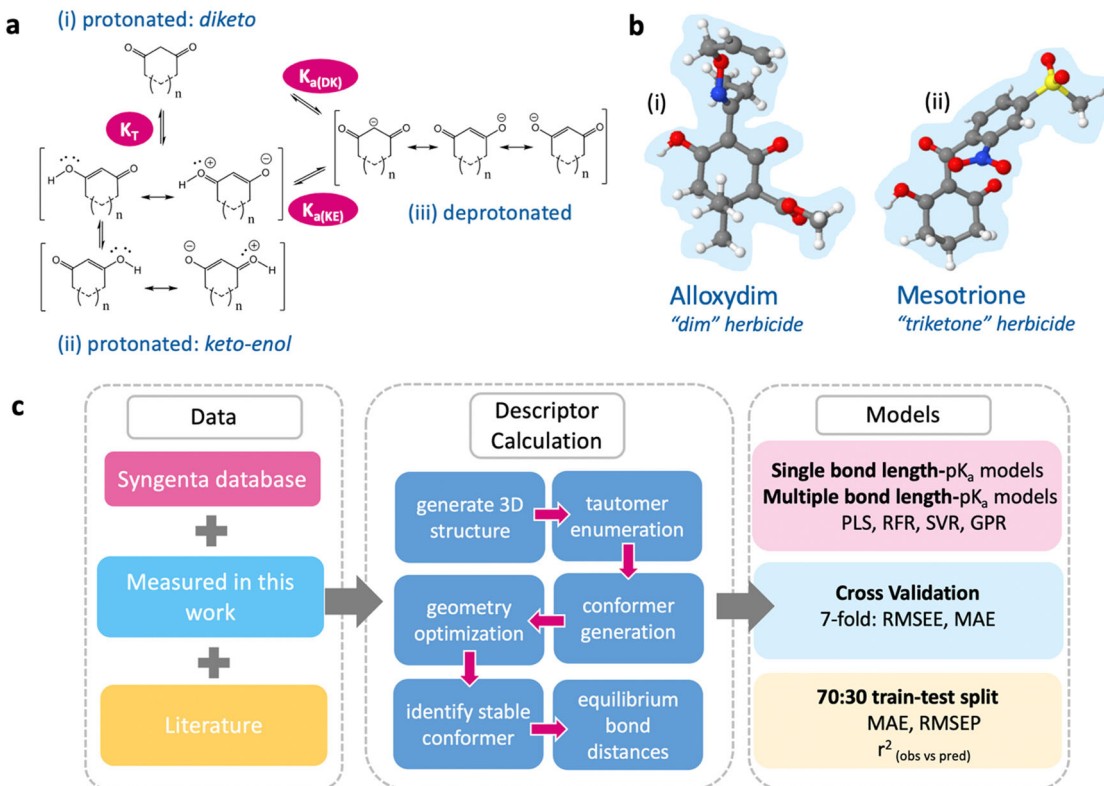

**Fig. 1 Structures of 1,3-diketone derivatives and schematic of our workflow. a** (i) The diketo form of a 1,3-dione, (ii) the resonance canonicals for the keto-enol form of 1,3-diones, and (iii) the resonance canonicals for the anionic state, where $n = 0$ or 1 if the ring is five- or six-membered, respectively. $K_T$ denotes the equilibrium constant between tautomeric states, $K_{a(DK)}$ denotes the dissociation equilibrium from the diketo state and $K_{a(KE)}$ the dissociation equilibrium from the keto-enol state. **b** (i) The global minimum geometry of Alloxydim, a 2-oxime herbicide and Mesotrione in the keto-enol *anti* state, (ii) a triketone herbicide. **c** The AIBL-$pK_a$ workflow implemented here for cyclic *β*-diketones.

other quantum chemically derived descriptors in the context of Quantitative Structure Property Relationship studies[26]. Most recently, our approach to pK$_a$ prediction, which uses only internuclear distances as descriptors, called AIBL-pK$_a$ (ab initio bond lengths), showed remarkably accurate prediction of acidity variation across congeneric series of guanidine-containing species[19] and sulfonamides[20]. The current work brings attention to the issue of pK$_a$ prediction for tautomerizable compounds and delivers an intuitive solution to this problem for 1,3-CHD and 1,3-CPD derivatives, which remain important scaffolds in pharmaceutical and agrochemical research.

## Results and discussion

**Scheme for model construction.** Our proposed method of predicting pK$_a$ values (Fig. 1c and Methods) makes use of equilibrium bond lengths from density functional theory calculations (B3LYP/6-311G(d,p) with the conductor-like polarizable continuum model or CPCM) as input features for regression models.

The full dataset of 71 compounds used in this work represent a wide variety of substituent types and patterns (generic structures and examples of dataset compounds are shown in Fig. 2a). After an initial analysis of the linear fit of each individual bond length, we investigate whether the use of multiple bond lengths as input features could provide an advantage in prediction accuracy and model applicability radius. For this task, we considered all subset combinations of the bonding distances of the fragment common to each species. We also compared a number of machine learning methods for their regression onto pK$_a$ values, namely, random forest regression (RFR), support vector regression (SVR), Gaussian process regression (GPR) as well as partial least squares (PLS). PLS[27] and SVR[28–30] have been implemented in the context of pK$_a$ prediction many times, using many different types of descriptors. A brief overview of the theory and method used for these approaches can be found in the Supplementary Methods section of the Supplementary Information (SI). Further

details and formalism for the validation metrics used in this work (r$^2$, RMSEE, MAE) can also be found in Supplementary Methods.

Through our analysis, we demonstrate that a powerful model may be constructed from simple linear regression of a single ab initio bond length, thereby potentially negating the need for the more complex approaches.

**Current approaches.** To exemplify the issues surrounding prediction for cyclic 1,3-diketones using existing empirical approaches, the commercial program by ChemAxon known as Marvin was used to estimate values for a series of 1,3-CHD and 1,3-CPD derivatives (**o1-o8**, **tk1-tk15** and **dk1-dk12** shown in Supplementary Table 1 of the SI). The Marvin program uses Gasteiger partial charges[30], polarizabilities and structure specific increments to predict pK$_a$ values using ionizable group specific regression equations[11]. The results are shown in Fig. 2b, where the orange diamonds denote experimental values, blue squares represent Marvin predictions without the option to "consider tautomers/resonance", while the magenta triangles are predictions made with this option. For the compounds in Fig. 2b where the blue and red points overlap, the program predicts the keto-enol state to be dominant, and delivers predictions that lie 0.8 units away from experimental values on average. However, for 60% of the compounds, the program predicts the diketo state to be dominant. For the series **o1-o8**, Marvin gives values of ~16 log units for 5 out of 8 species. For the remaining three compounds, **o1**, **o3** and **o7**, the program identifies the acidic proton (pK$_a$ ~ 17) at the 4 or 6 position on the 1,3-CHD ring.

The above results suggest that if accurate predictions are to be made (i.e. residual errors <1 pK$_a$ unit), then the user must have prior knowledge of the dominant keto-enol tautomeric form (blue squares in Fig. 2b). In the following sections we show that our method, which uses quantum chemically derived geometric descriptors, avoids such problems intrinsically. Despite the increased computation time compared with empirical

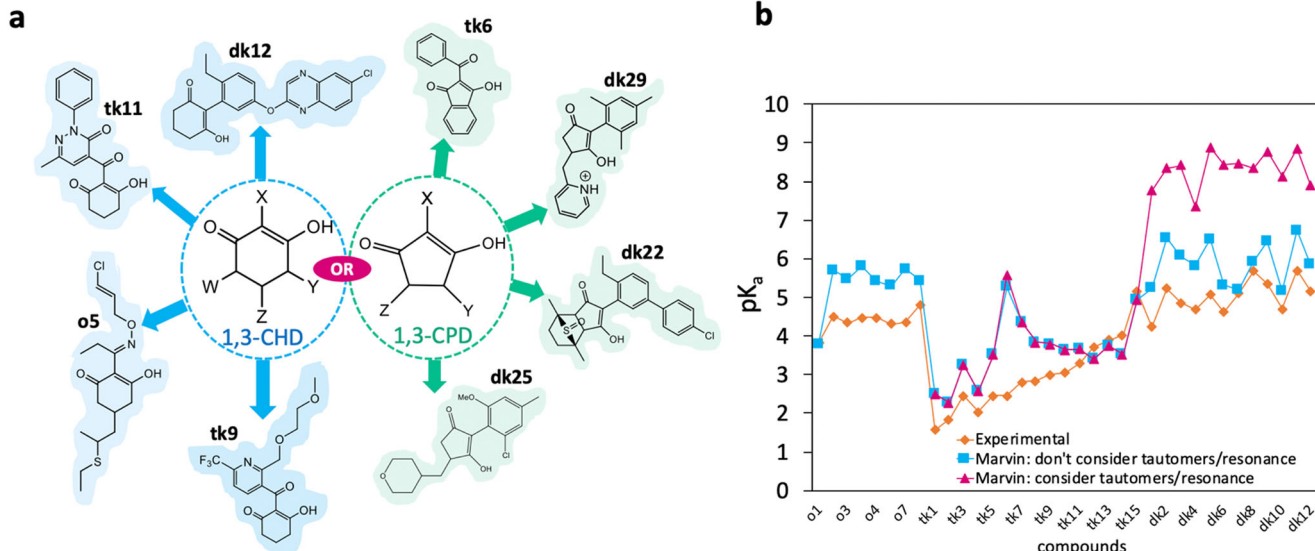

**Fig. 2 Exemplar cyclic diketone compounds studied in this work and the performance of Marvin versus Experiment. a** pK$_a$ data for compounds of the dataset used were procured from both Syngenta's database and literature sources. The pK$_a$ values of 17 compounds were also measured for the purpose of this work. Each compound either contains a 1,3-cyclohexanedione (1,3-CHD) or 1,3-cyclopentanedione (1,3-CPD) group, examples of which are shown in blue and green, respectively. Substituent variation occurs at 2, 4, 5 and/or 6 position on 1,3-CHD, and 2, 4 and/or 5 for 1,3-CPD. The full set of structures and experimental pK$_a$ values can be found in Supplementary Table 1 of the SI. **b** Experimental (orange) pK$_a$ values across the series **o1-o8**, **tk1-tk15** and **dk1-dk12**, are compared with Marvin predictions with the "consider tautomers/resonance" option (magenta) and without this option (blue). Values are in excess of 14 log units for the acidic proton at C2 (labelled for 1,3-cyclohexanedione) for **o2**, **o4**, **o5**, **o6** and **o8** and values for **o1**, **o3** and **o7** correspond to C4/C6.

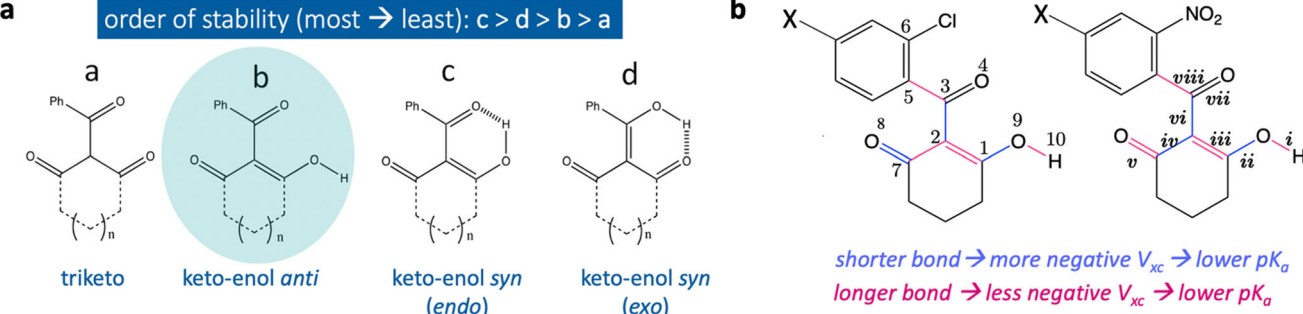

**Fig. 3 Tautomeric forms of cyclic triketones and the trends in bond length variation with pK$_a$ for compounds labelled tkn and tkc in this work. a** Tautomeric forms **a–d** considered for the triketone series **tkn1-tkn4** and **tkc1-tkc6**. All energies are listed in Supplementary Table 6 of the SI. **b** The trend in bond length variation and exchange-correlation (V$_{xc}$) energy of bonding interactions for **tkn1-tkn4** is consistent with delocalization of electrons across the whole endocyclic keto-enol fragment. Conversely, the variation in bond lengths for **tkc1-tkc6**, as well as the increased co-planarity of the keto-enol group, is indicative that there is more conjugation with the *exo*-carbonyl. Supplementary Table 7 of the SI lists bond lengths **i–v** and pK$_a$ values for the **b** tautomer.

approaches, AIBL avoids the need to compute pK$_a$ values for both protonation states. Moreover, descriptor calculations may be carried out only in the solvent phase using an implicit approach (CPCM).

**Identifying AIBL-pK$_a$ relationships for triketones.** The relationship between the structure and herbicidal activity of trike-tones (Fig. 3a) was first reported[31] by Lee and co-workers. One of the primary conclusions of that early work was that the *ortho*-substituent on the phenyl ring is a requirement for the compound's herbicidal activity. The authors also noted that compounds with more electron-withdrawing *para*-substituents required a lower dose to obtain a 50% weed-control rating across 7 variants of broad-leaf plants (the metric known as lethal dose 50, or LD$_{50}$). It was thereby deduced that a linear relationship exists between Hammett constants of *para*-substituents, log (LD$_{50}$) and pK$_a$. Therefore, a more electron-deficient benzene is associated with enhanced acidity and herbicidal activity[31]. As there is already evidence of a structure–property/activity relationship for these species, we took the set of 10 compounds from the work of Lee et al. as a starting point to assess the prevalence of AIBL-pK$_a$ relationships across available tautomeric states.

The identities, pK$_a$ values, equilibrium bond lengths and log (LD$_{50}$) values of the compounds studied by Lee et al. are shown in Supplementary Tables 2–5, labelled as **tkn1-tkn4** and **tkc1-tkc6**. All **tkn** species possess one 2-NO$_2$ group whereas each **tkc** species has a 2-Cl substituent (Fig. 3b). Across each subset the *para*-substituent varies. We find that the order of stability of each compound in their four lowest energy tautomer/conformations (Fig. 3a) is **c > d > b > a**. The triketo form **a** is ~9 kJ mol$^{-1}$ less stable than the (*endo*) keto-enol *anti* form **b**, which in turn is ~29 kJ mol$^{-1}$ less stable than the (*exo*) keto-enol *syn* form **d**. Although both **d** and **c** possess a stabilising intramolecular hydrogen bond, the most stable form is **c** by around 7 kJ mol$^{-1}$.

Experimental pK$_a$ values were regressed onto bond lengths **i–viii** (Fig. 3b) of the triketo or keto-enol fragment of tautomers **a–d** and the fit was assessed using r$^2$. For all tautomers **a–d**, there is a significant improvement in r$^2$ when the set is split into two subsets (r$^2$ generally 0.9 or above), with one group containing **tkn** derivatives and the other containing **tkc** substituted compounds. The slope for the **tkn** series is consistently 22% larger (i.e. steeper) than that of the **tkc** derivatives. We can interpret this steeper gradient as the resonance electron-withdrawing effect of the 2-NO$_2$ substituent heightening the *para*-substituent's electronic effect on dissociation propensity. The heightened acidity of the **tkn** compounds is also likely to be linked to the marked difference

in geometry between the two subsets. For the **tkc** series, the exo-carbonyl group is almost co-planar with the phenyl ring, whereas for the **tkn** series, the exo carbonyl is co-planar with the keto-enol moiety. In the latter orientation (of the **tkn** series), the orbital overlap allowing hydroxyl oxygen lone pair delocalisation across the keto-enol and exo-keto group is possible. It may be asserted that this increased conjugative effect would result in less delocalization between O and H atoms, a longer, weaker O–H bond and greater propensity for dissociation.

The bond lengths of the enol *anti*-conformer **b** exhibit the most strongly correlated relationships with pK$_a$ values (see Supplementary Tables 2–5). With the exception of O–H(**i**) and the exocyclic C=O(**vii**) bond lengths, all pairs of subsets **tkn** and **tkc** exhibit r$^2$ values above 0.90 ($q^2 > 0.9$ and RMSEE ~ 0.2). This is an interesting result, considering that **b** is not the most stable tautomer according to the ranking at B3LYP/6-311 G(d,p)/CPCM. It may be asserted that the emergence of stronger relationships between geometric features (bond lengths) and pK$_a$ using the *anti* keto-enol tautomer is indicative of its prevalence in solution. A thorough analysis using explicit solvation to explore this hypothesis is beyond the scope of this work. However, preference for this conformation could be linked to its increased propensity for dimerization and H-bonding to solvent molecules.

For both subsets, the trend in the bond variation of O–H (**i**), C–O (**ii**) and C=C (**iii**) with pK$_o$ is such that more acidic compounds have longer O–H and C=C bonds but shorter C–O distances. These observations therefore fit with the intuition that a longer, weaker O–H bond should exhibit an increased propensity for cleavage. Conversely, bonds C–C (**iv**) and C=O (**v**) are found to show opposing trends between each series (Fig. 3b).

The aim of this work is to derive a generally applicable model for compounds containing the diketone fragment. Therefore, we deemed it important to understand this disparity in C–C (**iv**) and C=O (**v**) bond length variation. To this end, we performed an interacting quantum atoms (IQA) analysis to partition the interaction energy between pairwise atoms A and B into $V_{xc}$(A, B) (exchange-correlation) and $V_{cl}$(A,B) (electrostatics). For further methodological and theoretical details of this approach see the Methods section.

By taking $V_{xc}$(A,B) as our dependent variable in place of bond distances, we can look at how the extent of delocalization of electrons between two topological atoms A and B changes with pK$_a$. In doing so, we find analogous relationships between $V_{xc}$(A,B) of bonds **i–v** and pK$_a$ values. Longer bonds exhibit less negative $V_{xc}$(A,B) values (i.e. there is less delocalization), and vice

versa (Fig. 3b). The trend in $V_{xc}(A,B)$ for bonds **i–v** across the keto-enol fragment of the **tkn** series is consistent with hydroxyl oxygen lone pair delocalization across the whole keto-enol fragment, akin to the resonance forms shown in Fig. 1a(ii). Conversely, for the **tkc** series this delocalization effect is not reflected in the distance variation of **iv** and **v**. Further discussion pertaining to the origin of the difference in bond and $V_{xc}$ variation with $pK_a$ between subsets can be found in Supplementary Methods. Overall, the discrepancy in AIBL-$pK_a$ trends with substituent type (Supplementary Note 2) suggests that, in the search for a bond that has a relationship with $pK_a$ over a wide variety of substituent patterns/types, it is logical to look to the enolic hydroxyl group, i.e. O–H (**i**), C–O (**ii**) and C=C (**iii**).

Due to the prevalence of well-correlated relationships between bonding distances and $pK_a$ for the keto-enol *anti*-conformation for **tkn1-tkn4** and **tkc1-tkc6**, this tautomeric form was used for all subsequent analysis on the remaining dataset. The bonds that are under investigation are those of the keto-enol fragment (**i–v** in Fig. 3b), which are common to all 1,3-CHD and 1,3-CPD compounds of the dataset. Selection of these specific bond lengths therefore allows us to construct one generally applicable model, rather than assembling many models for more specific sub-regions of chemical space.

**Single bond length models**. Our dataset of 71 compounds (Supplementary Table 1) consists of 46 triketones and diketones from Syngenta, plus an additional 9 diketones and 2 triketones measured for the purpose of this work (experimental details can be found in Supplementary Methods in the SI). A further 8 $pK_a$ values for Alloxydim analogues were also obtained from the literature (Supplementary Table 1). Due to a discrepancy between predicted and literature values, samples were procured and $pK_a$ values were re-measured for 7 of these 8 compounds. Literature values for 6 Tetracycline derivatives were also included. The full set was split into 70% training and 30% test set, i.e. 49:22 training to test set.

Table 1 lists internal, cross-validation and external validation statistics of each single bond length regression model (i.e. the typical AIBL approach). The values listed in Table 1 are found using a reduced training set, due to the removal of two outliers, **dk29** and **tk3**. The reason for the removal of these compounds will be discussed in the next section. The most active bond, i.e. the model exhibiting the highest $r^2$ and lowest RMSEE is the C–O (**ii**)

bond (0.72 and 0.57, respectively). We note that these values are somewhat less impressive than the threshold values used to mark the presence of an active bond in our other case studies (~0.90 for $r^2$ and ~0.3 for RMSEE). This decrease in goodness of fit can be attributed to the higher structural diversity of the set: the model covers 5- and 6-membered rings, compounds with substitution at the 2, 4 and 6 position of the 1,3-CHD fragment and compounds containing more than one ionizable group.

Nonetheless, the error metrics for the C–O model ($pK_a = 93.381 \cdot r(CO) - 127.71$) used on the external test set indicate a high level of prediction accuracy and consistency across a diverse array of analogues; the MAE and standard deviation of absolute errors for the test set are both 0.24. No C–O model errors exceed 1 $pK_a$ unit and only 2 out of 22 exceed 0.5 log units (**tk1** = +0.92, **dk8** = −0.77). The nature of bond length variation across the 47 training compounds matches that of the **tkn/tkc** series for O–H (**i**), C–O (**ii**) and C=C (**iii**).

**Outliers**. Two species were found to have residual errors exceeding 1.5 log units for 4 out of 5 bonds. One outlier is **dk29**, a 1,3-CPD derivative with a $CH_2$-2-pyridyl group at the 4-position. The $pK_a$ value of 5.78 listed for this species was identified as the $pK_a$ for dissociation of the 2-pyridyl group, rather than the keto-enol fragment (pyridine itself has a $pK_a$ of 5.23). The other incongruous data point corresponds to **tk3**, which has a fourth keto group at the 5-position of the 1,3-CHD ring, a feature that is also present in compounds **tk1** and **tk4**. The C–O bond distances of these three compounds sit below the trend line for the rest of the set, with an $r^2$ value of 1 for a linear fit, i.e., compounds with the 5-C=O structural motif in common form their own high-correlation subset. More accurate predictions for compounds such as **tk1** (error = +0.92) could therefore be made using the equation of this line as a new model, rather than the original C–O model. Both compounds were removed from subsequent analysis.

**Other regression approaches**. Table 2 shows the 7-fold CV and external validation statistics for optimal models. These were derived using PLS (4 bonds), RFR (3 bonds), SVR [linear] (2 bonds), SVR [RBF] (3 bonds) and GPR [RBF] (3 bonds) using feature selection based on minimization of the 7-fold RMSEE of the training set. The 7-fold RMSEE for each of the 31 combinations/subsets are compared in Fig. 4a (the Model ID list is shown in Supplementary Table 8, the full list of statistics for each model is shown in Supplementary Tables 9–13 and predictions are shown in Table 3). The optimal model for each method was then used to predict test set $pK_a$ values.

Overall, all optimal models for each method include C–O as an input feature. The lowest 7-fold CV MAE and RMSEE correspond to the GPR model using a radial basis function kernel, which uses C–O, C–C and C=O as input features (MAE = 0.30, RMSEE = 0.39). However, this same GPR model also delivers the least accurate predictions for the 22 compounds of the external test set with an RMSEP of 0.59 and a MAE of 0.43, possibly indicative of overfitting to the training set data. Overall, SVR[RBF] using C–O, C–C and C=O returns the lowest MAE and RMSEP for the test set (0.29 and 0.36, respectively) and is consistent in its accuracy (s.d. = 0.22). However, PLS using C–O, C=C, C–C and C=O also performs similarly well (MAE = 0.31, RMSEP = 0.36) and exhibits the lowest standard deviation of absolute errors (s.d. = 0.19). There is one consistently large error across every model, corresponding to the predicted value for **tk1**. This compound shows an average error across all models of −1.21, with the lowest error exhibited by the PLS model (−0.72) and the largest for GPR[RBF] (−1.60). This compound was previously identified as belonging to a new subset of 5-C=O

**Table 1 Summary of the results for the typical AIBL ordinary least squares approach.**

| Metric | O–H (i) | C–O (ii) | C=C (iii) | C–C (iv) | C=O (v) |
|---|---|---|---|---|---|
| Slope (+/−) | − | + | × | × | × |
| $r^2$ (train) | 0.56 | 0.72 | 0.38 | 0.15 | 0.38 |
| MAE (7-fold CV) (train) | 0.60 | 0.41 | 0.65 | 0.88 | 0.73 |
| RMSEE (7-fold CV) (train) | 0.75 | 0.57 | 0.89 | 1.10 | 0.90 |
| MAE (test) | 0.31 | 0.24 | 0.43 | 0.67 | 0.56 |
| RMSEP (test) | 0.41 | 0.34 | 0.58 | 0.86 | 0.69 |
| s.d. (test) | 0.28 | 0.24 | 0.40 | 0.55 | 0.41 |
| $r^2$ obs vs pred (test) | 0.90 | 0.92 | 0.69 | 0.66 | 0.20 |

(Upper) Statistics for the single bond length models obtained via ordinary least squares regression. The row labelled "slope" features a "+" sign for a positive slope (i.e. $pK_a$ increases with increasing bond distance), and a "−" sign to denote a negative slope (i.e. $pK_a$ decreases with increasing bond distance). The squared correlation coefficient is not significant enough ("×") to assign a slope direction for **iiii**, **iv** and **v**.

**Table 2 Summary of the results for optimal feature choice using PLS, RFR, SVR with linear and RBF kernels, and GPR with the RBF kernel.**

| Property/Metric | Marvin | PLS | RFR | SVR [linear] | SVR [RBF] | GPR [RBF] |
|---|---|---|---|---|---|---|
| Features used | – | C–O, C=C, C–C, C=O | C–O, C–C, C=O | C–O, C=O | C–O, C–C, C=O | C–O, C–C C=O |
| | | | Max depth = 6 | C = 1000 | C = 1000 | $\ell = -8.21$, $-6.150$, $-12.851$ |
| Hyperparameters | – | LV = 3 | | | $\varepsilon = 0.1$ | |
| | | | $n_{est} = 25$ | $\varepsilon = 0.01$ | $\gamma = 5$ | |
| MAE (7-fold CV) (train) | – | 0.41 | 0.46 | 0.43 | 0.40 | 0.30 |
| RMSEE (7-fold CV) (train) | – | 0.53 | 0.57 | 0.57 | 0.53 | 0.39 |
| MAE (test) | 1.21 (4.70) | 0.31 | 0.39 | 0.29 | 0.29 | 0.43 |
| RMSEP (test) | 1.63 (6.32) | 0.36 | 0.49 | 0.40 | 0.36 | 0.59 |
| s.d. (test) | 1.12 (4.32) | 0.19 | 0.31 | 0.28 | 0.22 | 0.36 |
| $r^2$ obs vs pred (test) | 0.61 (0.55) | 0.86 | 0.74 | 0.90 | 0.86 | 0.67 |

The "Marvin" column corresponds to statistics for predictions made without considering tautomers/resonance (without parentheses), and the values in parentheses correspond to the predictions made with consideration of tautomers/resonance. The "features used" row lists the combination of features that minimized the RMSEE of the training set for each method. These features were subsequently used in the model used to predict for test set compounds. The row labelled "hyperparameters" lists the values obtained through minimization of RMSEE of the training set during 7-fold cross-validation (RFR and SVR). For PLS the number of latent variables (LV) was varied up to the number of features and the final number chosen on the basis of minimizing the RMSEE of the training set, which is also shown. For the GPR model, feature selection as carried out using 7-fold validation of each combination/subset of features using the training set and 100 restarts were used to locate the global maximum log likelihood of the $y$-values. The MAE, RMSEP, standard deviation of absolute errors (s.d.) and $r^2$ of observed vs predicted values are shown for the test set.

containing compounds, along with **tk3** and **tk4** for the C–O model, and may therefore be considered to be on the edge region of the domain of applicability for the model.

The comparable accuracy of the single bond length C–O model for the test set, with respect to more complex regression methods using more input features is a remarkable result, given the simplicity of the approach. This result also validates our previous work, in which models using multiple input features were deemed unnecessary given the strength of the correlation for individual bond distances.

**Marvin**. A comparison between error metrics for all models shows significant improvement compared with Marvin (Figs. 4b, c), either with or without consideration of tautomer/resonance. Furthermore, AIBL provides predicted values that correctly suggest the dominant microstate at pH 7 is the enolate, i.e. the ionized form. After tautomer enumeration and selection, Marvin's $pK_a$ values predict that 15 out of 22 compounds would be >50% unionized at this pH. However, this result is reduced to only two incorrectly assigned microstates when the keto-enol form is used explicitly. All experimental and predicted values can be found in Table 3.

**Correction of experimental value for Profoxydim**. Experimental $pK_a$ data were initially procured from literature sources for the series of "dim" herbicides used in this work (Supplementary Note 1). Upon performing the fits for the single bond length models, the residual error for Profoxydim (Fig. 4d) using the literature $pK_a$ value of 5.91 was found to be anomalously high, at +1.30 units. Marvin predicts the $pK_a$ of the enolic hydroxy group to be 5.44, i.e. very close to this experimental value.

Due to the excellent accuracy observed for species **o1**-**o7** (residuals < 0.50), we decided to re-measure all $pK_a$ values. Seven of the eight compounds (all except Clethodim) were procured and re-measured using the UV-metric method (see Supplementary Methods for details). Excellent agreement was found between old and new values for all compounds but Profoxydim, for which a value of 4.82 was found. This new value lies only 0.22 units from our original prediction (4.61), yet it lies ~1.10 log units from the literature value. Therefore, we demonstrate the power of the AIBL

approach to check internal consistency of $pK_a$ values for a given congeneric series. Structures and predictions for all dim herbicides can be found in Supplementary Fig. 2 of the SI.

**Tetracyclines**. Aside from tautomerism, one of the more complex issues in the field of $pK_a$ prediction is the estimation of values for multiprotic compounds. Two of the species of our dataset contain a secondary ionizable group (**dk26** and **dk29**, 2-pyridyl, $pK_a = $ ~5). In recent work we have demonstrated that prediction for a specific ionizable group may be performed by using the relevant microstate to the dissociation of interest. Therefore, in the case of **dk26** and **dk29**, we performed all calculations on the cationic form of the 2-pyridyl group. To showcase the applicability of the AIBL model derived here in the context of larger multiprotic compounds, 6 tetracycline derivatives were included. For the correct microstate (the neutral state) of each species the most stable form is analogous to the keto-enol *syn* **c** conformation. The *anti*-conformation was constructed by manual rotation of the $C^2$–$C^1$–$O^9$–$H^{10}$ (Fig. 3b) torsional angle from this form. For **tet1**, **tet3**, **tet5** and **tet6** of the training set, residual errors from the C–O model are below 0.1 log unit in all cases. For the test set compounds, predictions for **tet2** and **tet4** also lie within 0.1 log units. Use of Marvin with consideration of tautomers on this occasion identifies the keto-enol state as the relevant tautomeric form, delivering predictions of 2.83, 2.63, 2.55, 2.92, 2.84 and 2.51, for **tet1**–**tet6**, respectively, whereas experimental values are 3.35, 3.48, 3.25, 3.50, 3.53 and 3.30, respectively. Therefore, despite making the prediction using the correct tautomer, there is a distinct prediction bias towards higher acidity for the enolic hydroxy group for these compounds. Structures and predictions for tetracyclines can be found in Supplementary Fig. 3 of the SI.

**Future application of AIBL**. The poorer performance of Marvin, as illustrated by Figs. 4b, c, can most likely be partly attributed to a lack of coverage of this type of compound (cyclic 1,3-diketones) in their training dataset. The predicted preference of the diketo state of many test compounds can also likely be attributed to the lack of knowledge on relative tautomeric stability, as previously pointed out by Connolly. The results in Fig. 4 illustrate the excellent performance of the C–O AIBL-$pK_a$ model in predicting

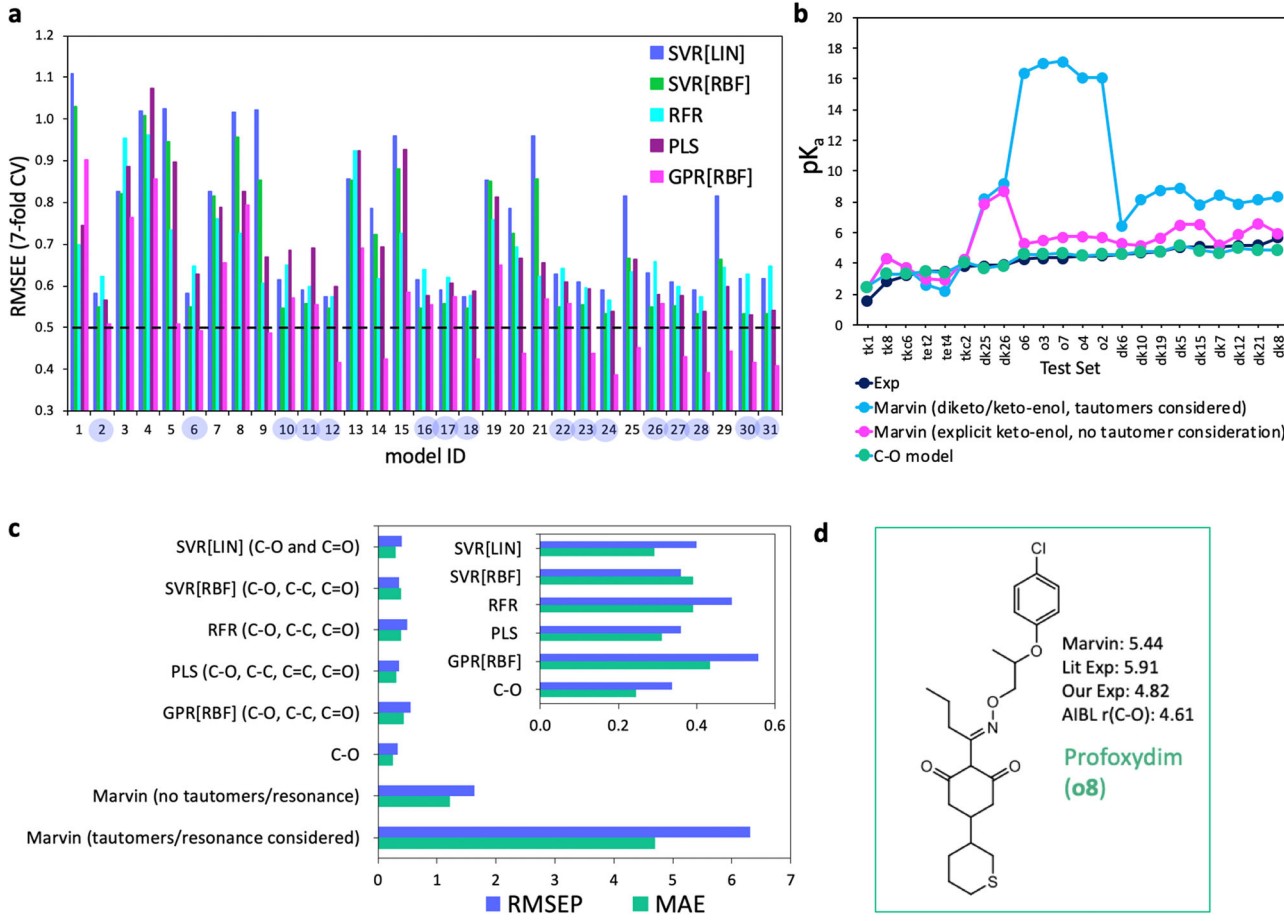

**Fig. 4 Performance of each regression method tested in this work using bond lengths as input features compared with results obtained using Marvin. a** The 7-fold RMSEE for each model tested, for each method, where "Model ID" corresponds to one of 31 combinations of features out of the 5 bonds **i–v** chosen for consideration (see Supplementary Table 8 for the full list). The C–O, **ii** bond is used as a feature for the Model ID numbers shaded in blue. **b** Experimental $pK_a$ variation across the test set (dark blue), along with Marvin predictions using the diketo state with tautomer consideration turned on (blue), and using the keto-enol state with tautomer consideration turned off (magenta), as well as the AIBL-$pK_a$ C–O bond model (green). **c** Root-mean squared error of prediction for the test set (RMSEP, blue) and mean absolute error for the test set (MAE, green) for each method of prediction. Marvin predictions are removed for the plot shown in the inlay, so that AIBL models can be compared. **d** The structure of Profoxydim, for which the literature experimental $pK_a$ value (5.91) and Marvin's prediction (5.44, tautomer/resonance not considered, keto-enol form used) deviated significantly from our prediction. The new experimental value of 4.82, measured in this work matches our initial prediction more closely.

the $pK_a$ variation across the series. Furthermore, we show that the accuracy is such that we can correct experimental values. We assert that a powerful future application of the AIBL approach is a method of fleshing out areas of chemical space that are sparse in the experimental $pK_a$ databases of empirical predictors, such as Marvin. Once a model has been set up with existing experimental data, hypothetical compounds with a variety of substituents can be assembled and their $pK_a$ values predicted and added to the training set. Therefore, the empirical approach is calibrated using the highly accurate AIBL approach, whilst still maintaining user-friendly computational speed.

We have shown bonding distances to be an intuitive and powerful descriptor of ionization propensity for much of 1,3-CHD and 1,3-CPD space. Due to the use of quantum chemically derived descriptors, the dominant tautomeric state is easily identified as the keto-enol form, from which chemically meaningful relationships are derived; a longer O–H and a shorter C–O bond are generally indicative of a species with heightened acidity compared with the parent compound. A simple but accurate AIBL-$pK_a$ method is proposed and validated; good results are derived using only simple linear

regression of $pK_a$ onto C–O bond distances, which is shown to be applicable to a diverse array of analogues. For the test set, this simple model is found to outperform regression using various approaches and multiple bond lengths relevant to the dissociation at the keto-enol ionizable group. Furthermore, the method is applicable to multiprotic compounds, which along with tautomerizable species, represent one of the most challenging areas of $pK_a$ prediction. All of the models developed showed superior accuracy compared with the industry standard, represented by the program Marvin, for which the user must have prior knowledge of the dominant tautomeric form. At present, there is still a time/cost barrier to feasible use of quantum chemical QSPR methods in large scale screening studies. However, this work suggests that the inclusion of some description of electrons and their distribution (via a highly populated geometric representation of molecules), provides a significant advantage in terms of prediction accuracy over an approach (Marvin) that does not describe a compound quantum mechanically. Thanks to AIBL predictions, we also amend the literature experimental value for Profoxydim, which is corrected from a previous value 5.91 to a new value of 4.82. Based on the

**Table 3 Experimental pK$_a$ values and predictions for each method tested in this work, for the test set compounds.**

| ID | Exp | C-O | PLS | RFR | SVR[LIN] | SVR[RBF] | GPR[RBF] | Marvin (taut) | Marvin (no taut) |
|----|-----|-----|-----|-----|----------|----------|----------|---------------|------------------|
| tk1 | 1.56 | 2.48 | 2.28 | 2.71 | 2.66 | 2.34 | 3.16 | 2.44 | 2.44 |
| tk8 | 2.84 | 3.33 | 3.45 | 2.78 | 3.40 | 3.39 | 2.88 | 4.35 | 4.35 |
| tkc6 | 3.20 | 3.34 | 3.45 | 2.79 | 3.41 | 3.39 | 2.87 | 3.73 | 3.73 |
| tet2 | 3.48 | 3.52 | 3.63 | 2.83 | 3.58 | 3.54 | 3.01 | 2.63 | 2.99 |
| tet4 | 3.50 | 3.42 | 3.45 | 2.91 | 3.48 | 3.40 | 2.99 | 2.19 | 2.95 |
| tkc2 | 3.83 | 4.08 | 4.17 | 4.41 | 4.05 | 4.09 | 4.07 | 4.29 | 4.29 |
| dk25 | 3.85 | 3.69 | 3.29 | 2.86 | 3.72 | 3.26 | 3.21 | 8.17 | 7.87 |
| dk26 | 3.92 | 3.85 | 3.59 | 3.35 | 3.85 | 3.55 | 4.27 | 9.18 | 8.70 |
| o6 | 4.30 | 4.64 | 4.75 | 4.82 | 4.52 | 4.69 | 5.11 | 16.34 | 5.30 |
| o3 | 4.34 | 4.64 | 4.74 | 4.73 | 4.52 | 4.69 | 5.09 | 17.00 | 5.47 |
| o7 | 4.35 | 4.70 | 4.83 | 4.92 | 4.56 | 4.76 | 5.14 | 17.15 | 5.73 |
| o4 | 4.47 | 4.53 | 4.66 | 4.58 | 4.43 | 4.58 | 5.05 | 16.09 | 5.79 |
| o2 | 4.51 | 4.63 | 4.73 | 4.79 | 4.52 | 4.67 | 5.08 | 16.06 | 5.70 |
| dk6 | 4.62 | 4.62 | 4.75 | 4.56 | 4.50 | 4.63 | 4.57 | 6.44 | 5.31 |
| dk10 | 4.71 | 4.79 | 4.83 | 4.36 | 4.62 | 4.76 | 4.35 | 8.14 | 5.17 |
| dk19 | 4.76 | 4.77 | 4.63 | 4.75 | 4.66 | 4.66 | 4.76 | 8.76 | 5.66 |
| dk5 | 5.06 | 5.17 | 5.26 | 5.17 | 4.94 | 5.21 | 5.09 | 8.90 | 6.50 |
| dk15 | 5.08 | 4.87 | 4.81 | 4.98 | 4.74 | 4.82 | 4.98 | 7.83 | 6.54 |
| dk7 | 5.10 | 4.69 | 4.82 | 4.89 | 4.55 | 4.71 | 4.73 | 8.46 | 5.19 |
| dk12 | 5.16 | 5.02 | 5.11 | 5.19 | 4.80 | 5.05 | 4.84 | 7.88 | 5.85 |
| dk21 | 5.19 | 4.89 | 4.87 | 4.98 | 4.77 | 4.84 | 5.02 | 8.11 | 6.61 |
| dk8 | 5.69 | 4.92 | 5.18 | 5.16 | 4.76 | 5.01 | 5.24 | 8.35 | 5.93 |

work shown here, and on previous results, we propose that AIBL-pK$_a$ is applicable to any tautomerizable congener series, given that pK$_a$ data exist for model calibration.

## Methods

**Data**. Structures and pK$_a$ values with references are given in Supplementary Table 1 for all compounds studied in this work. Equilibrium bond lengths for the most stable geometries identified are listed in Supplementary Table 7.

The pK$_a$ data for the compounds investigated in this work have been procured from various sources. Sixteen triketones, labelled **tk-1** to **tk-15**, **tk18** and **tk19** were procured from the Syngenta and are analogues of the herbicide Mesotrione. A further 20 diketone compounds were procured from Syngenta, which are labelled as **dk-1** to **dk-12** and **dk22** to **dk29**. These values were obtained using the UV-vis metric approach with a Sirius T3 instrument at standard conditions (see Supplementary Methods in the SI for more details). A set of 10 compounds of triketone (**tk**) type labelled in as **tkn1-tkn4** and **tkc1-tkc6** were taken from the work[32] of Lee et al. Samples of 11 diketones (**dk**), labelled **dk-13** to **dk-21**, **tk16** and **tk17** have been procured and measured for the purpose of this work, using the potentiometric method with a Sirius T3 instrument at standard conditions. Finally, literature values were procured for 8 "dim" herbicides Alloxydim, Cycloxydim, Butroxydim, Clethodim, Sethoxydim, Tepraloxydim, Tralkoxydim and Profoxydim were procured, samples were purchased for all except Clethodim (due to unavailability) and pK$_a$ measurements were taken using the same apparatus and experimental procedure as described above and in Supplementary Methods. Literature values for 6 tetracycline derivatives (**tet1**–**tet6**) were obtained from literature sources.

**Quantum chemical calculations**. An ensemble of 15 conformers were generated for each tautomeric form of each compound **tkn1-tkn4** and **tkc1-tkc6** using the conformer generator plug-in within the Marvin program. Geometry optimization and frequency calculations were then performed using B3LYP/6-311G(d,p) with CPCM implicit solvation for each conformer of every ensemble using GAUSSIAN09[33]. Conformers were ranked according to internal energy and the most stable species was taken as the global minimum. For the *anti* and *syn* conformers of the keto-enol state, an input geometry for the higher energy *anti*-conformation was manually generated by rotating the orientation of the O–H bond of the *syn* conformer by 180°. This process of generating the keto-enol *anti* state[15,16,18–22] was repeated for the remaining 61 species.

**IQA calculations**. The extent of electronic delocalization between two atoms can be calculated within the context of a topological energy decomposition framework called interacting quantum atoms (IQA). Originating from the quantum theory of atoms in molecules[33] (QTAIM), IQA has been used to analyze a variety of chemical phenomena[34–38]. By decomposing the total energy of a system into intra- and interatomic terms, we derive the exchange-correlation potential energy $V_{xc}$, which is the sum of the exchange energy $V_x$, and the correlation energy $V_c$. The former

term usually dominates and denotes the Fock-Dirac exchange, which describes the ever-reducing probability of finding two electrons of the same spin close to one another (i.e. the Fermi hole). The latter term is associated with the Coulomb hole and the electrostatic repulsion between electrons. The absolute value of $V_{xc}$ evaluated between two atoms can be taken as the extent delocalization of electrons between them and so can be interpreted as a measure of covalency. These values were obtained by the AIMAll program[39] (version 14), using DFT-compatible IQA partitioning, and using default parameters on wavefunctions obtained at the B3LYP/6-311G(d,p) level using CPCM.

**Models**. For more details of regression methods implemented in this work see Supplementary Methods in the SI. Model training and error evaluation were performed using scikit-learn[40]. Initially, ordinary least squares (OLS) regression of single bond distances and pK$_a$, and validation was performed using r$^2$ and 7-fold CV RMSEE and MAE to assess the linear relationships between bond lengths and pK$_a$. A random 70:30 split of training set to external test set was then performed (i.e. training set = 49, test set = 22). We compared the results of using more than one bond length of the keto-enol fragment using support vector regression (SVR) with a linear and radial basis function (RBF) kernel, random forest regression (RFR), partial least squares (PLS) and Gaussian process regression (GPR) with an RBF kernel. We also compared our test set prediction errors results with those obtained using the program Marvin. Each model was evaluated using error-based metrics, mean absolute error (MAE), standard deviation of absolute errors (s.d.), root-mean-squared error (RMSEP) and the r$^2$ of observed vs predicted values. An overview of the AIBL workflow used in the context of cyclic β-diketones is shown in Fig. 1c.

The optimal hyperparameters for the SVR models, C, $\varepsilon$ (and $\gamma$ for the RBF kernel) and RFR (number of estimators $n_{est}$, maximum depth) were found in each case by applying a grid search (GridSearchCV in scikit-learn). The final hyperparameter values were chosen to minimize a 7-fold cross-validation RMSEE.

The GPR model was implemented in python using the GPR package called George. The squared exponential (SE) kernel, or RBF, was used to set up the GPR models with a unique length scale (hyperparameter) for each dimension, also known as the automatic relevance determination kernel of the SE-ARD,

$$SE - ARD(x, x') = \exp\left(-\frac{1}{2}\sum_{d=1}^{N}\frac{|x - x'|^2}{\ell^2}\right) \quad (1)$$

The hyperparameters for this kernel were found by maximizing the log-likelihood function using the training set. The implementation for this used the gradient descent BFGS algorithm (implemented by scipy) on the negative gradient of the log-likelihood function (therefore finding the maximum of the function). As there can be many local maxima, the optimizer was restarted with random weights 100 times in an attempt to find the global maximum.

## Data availability

All data analysed during this study are included in this published article (and its Supplementary Information).

## Code availability

The exact code is not provided given that it was written using methods from sci-kit learn (v.0.20.1) and George (v.0.3.1) libraries, which are freely available. Optimal hyperparameters for each method have been provided in the Supplementary Information and are otherwise set to default.

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

## Acknowledgements

P.L.A.P. thanks the EPSRC for Fellowship funding (EP/K005472), while P.L.A.P. and B.A.C. thank the BBSRC for funding her PhD studentship under the "iCASE" award BB/L016788/1 (with a contribution from Syngenta Ltd) and for funding a subsequent postdoc with Impact Acceleration funding (IAA_105) (with a contribution of Lhasa Ltd). C.D. thanks the Ministerio de Ciencia, Innovación y Universidades (MCIU/AEI/FEDER, UE; grant RTI2018-093940-B-I00).

## Author contributions

B.C. performed all computational work and analysis aided by P.L.A.P. who oversaw the entire study. N.K. and T.F. procured experimental pK$_a$ data for **tk1-tk15**, **tk18** and **tk19**, **dk1-dk12**, **dk22-dk29**. M.B. performed experimental pK$_a$ measurements for compounds **o1-o8** and C.D. performed experimental pK$_a$ measurements for **dk13-dk21**, **tk16** and **tk17**. B.C. prepared the paper and SI, which were reviewed by all authors.

## Competing interests

The authors declare no competing interests.
