## [Peer Review File · Communications Chemistry]

REVIEWERS' COMMENTS:

Reviewer #1 (Remarks to the Author):

Review of the manuscript COMMSCHEM-19-0342-T:

"Solving the Problem of Aqueous pKa Prediction for Tautomerizable Compounds Using Equilibrium Bond Lengths"

The authors have developed a methodology for predicting aqueous pKa values (AIBL-pKa) that they have used in several publications applied to different families of compounds. This methodology is used here to predict the aqueous pKa of 1,3-cyclohexanedione and 1,3-cyclopentanedione derivatives (71 compounds were considered), which are involved in tautomeric equilibria.

The AIBL-pKa methodology uses B3LYP-CPCM/6-311G(d,p) bond lengths as input features for linear regression models. Based on the current and previous results, the authors propose that this methodology could be applicable to any tautomerizable congener series if experimental pKa data exists for model calibration.

This well written and organized paper reports both theoretical and experimental results. The volume of the work reported is quite large and the work is done with rigour. Discrepancies between the AIBL pKa predictions and previous experimental results guided new experimental determinations that confirmed the theoretical predictions. This work is of relevance to others in the community.

Some questions and minor suggestions follow:

1. I find it useful to be able to see the structures of alloxydim and mesotrione (shown in Fig. 1B) in the format used to indicate the structures of Fig. 1A.
2. A small typo (page 3, 1st paragraph, 2nd last line): "... containing the 1,3-CHD or group ..."
3. Have the authors considered exploring other more recent continuum solvation methods (e.g., SMD) in their AIBL-pKa methodology using bond distances?
4. I think it would be relevant to cite previous publications in which molecular descriptors are used to predict aqueous pKa values. For example: J. Mol. Struct. (THEOCHEM) 2004, 684, 121; J. Phys. Chem. B, 2006, 110, 20546.
5. I think that the main linear regression equation derived using C-O bond distances should be explicitly reported in the paper.

Reviewer #2 (Remarks to the Author):

The authors tackle a nagging problem in pKa prediction, that of handling compounds with tautomeric forms. They propose that in fact a simple approach using calculated bond lengths gives quite impressive results, even compared to seemingly more advanced and complicated methods. They back up this claim with an impressive study, which contains a great deal of information, much of it relegated to the Supplemental Information. This latter fact may make it difficult reading for interested colleagues. It might be beneficial if the authors could perhaps put a table comparing estimated and experimental pKas in the main text, and perhaps a plot of the same data. Nonetheless this is a very impressive body of work, that will be of considerable interest to workers

in this area. In later work it would be of interest to have the authors expand further on just why this simple method works so well.

Reviewer #3 (Remarks to the Author):

In this paper, pKa is calculated for herbicide / therapeutic derivatives of 1,3-cyclohexanedione and 1,3-cyclopentanedione by utilising the linear relationship between bond length and pKa obtained from quantum chemical calculations. This is an interesting research and should be published in a general chemistry journal. However, there seem to be various opinions about whether or not to publish this paper in this journal, Comm. Chem.

One of the reasons is that the author have published a several papers by utilising the same relationship between the bond length and pKa (ref. 16-20). The target molecular species are diverse and differ in each paper, but they use essentially the same or similar computational procedure. The results obtained in the present study also seem to be of sufficient value for the molecules targeted for calculation. However, from the perspective of methodology novelty, it can be regarded as a routine work.

In addition, it should be pointed out that the relationship between bond length and pKa has been known for a long time. The following papers are examples;

A. J. Kirby, *Adv. Phys. Org. Chem.* 1994, 29, 87-183.

A. J. Green, J. Giordano, J. M. White, *Aust. J. Chem.* 2000, 53, 285-292.

J. E. Davies, N. L. Doltsinis, A. J. Kirby, C. D. Roussev, M. Sprik, *J. Am. Chem. Soc.* 2002, 124, 6594-6599

Moreover, the following might be in a similar position;

R. Fujiki et al. *Phys. Chem. Chem. Phys.*, 2018, 20, 27272-27279.

and their related works.

They are more or less related to linear free energy relationships. Indeed, the title of one of the authors' paper (Ref 19)

Linear Free-Energy Relationships between a Single Gas-Phase Ab Initio Equilibrium Bond Length and Experimental pK(a) Values in Aqueous Solution, indicating that this point is well recognized by the authors.

Roughly speaking, the direction of research progress based on computational chemistry can be divided into deductive-type based on physical chemistry and inductive-type on information/chemoinformatics. The importance of the latter seems to increase more and more from a pragmatic point of view. However, as described above, since this linear relationship is already well known, it is desired to systematically improve the prediction accuracy based on new ideas, for example, in information science. At the same time, one might feel that this linear relationship is obvious or self-evident. But the reviewer does not agree this. In view of the difficulty of evaluating free energy in solution, it is not easy to explain this linear relationship sufficiently logically from a physicochemical standpoint. It is highly expected to approach from at least one of these two positions, physical chemistry or informatics.

REVIEWERS' COMMENTS:

Review of the manuscript COMMSCHEM-19-0342-T:

"Solving the Problem of Aqueous pKa Prediction for Tautomerizable Compounds Using Equilibrium Bond Lengths"

Reviewer #1 (Remarks to the Author):

The authors have developed a methodology for predicting aqueous pKa values (AIBL-pKa) that they have used in several publications applied to different families of compounds. This methodology is used here to predict the aqueous pKa of 1,3-cyclohexanedione and 1,3-cyclopentanedione derivatives (71 compounds were considered), which are involved in tautomeric equilibria.

The AIBL-pKa methodology uses B3LYP-CPCM/6-311G(d,p) bond lengths as input features for linear regression models. Based on the current and previous results, the authors propose that this methodology could be applicable to any tautomerizable congener series if experimental pKa data exists for model calibration.

This well written and organized paper reports both theoretical and experimental results. The volume of the work reported is quite large and the work is done with rigour. Discrepancies between the AIBL pKa predictions and previous experimental results guided new experimental determinations that confirmed the theoretical predictions. This work is of relevance to others in the community.

Some questions and minor suggestions follow:

1. I find it useful to be able to see the structures of alloxydim and mesotrione (shown in Fig. 1B) in the format used to indicate the structures of Fig. 1A.

Supplementary Figure 1 has been added.

2. A small typo (page 3, 1st paragraph, 2nd last line): "... containing the 1,3-CHD or group ..."

Corrected.

3. Have the authors considered exploring other more recent continuum solvation methods (e.g., SMD) in their AIBL-pKa methodology using bond distances?

At this point we have not, but future work could consider a review of implicit solvent choice. The method was shown to work previously in the gas phase, we expect it to only improve with better solvent description.

4. I think it would be relevant to cite previous publications in which molecular descriptors are used to predict aqueous pKa values. For example: J. Mol. Struct. (THEOCHEM) 2004, 684, 121; J. Phys. Chem. B, 2006, 110, 20546.

Both papers have been added as references, thank you. However, the second paper does not use molecular descriptors.

5. I think that the main linear regression equation derived using C-O bond distances should be explicitly reported in the paper.

This is now included on page 9.

Reviewer #2 (Remarks to the Author):

The authors tackle a nagging problem in pKa prediction, that of handling compounds with tautomeric forms. They propose that in fact a simple approach using calculated bond lengths gives quite impressive results, even compared to seemingly more advanced and complicated methods. They back up this claim with an impressive study, which contains a great deal of information, much of it relegated to the Supplemental Information. This latter fact may make it difficult reading for interested colleagues. It might be beneficial if the authors could perhaps put a table comparing estimated and experimental pKas in the main text, and perhaps a plot of the same data.

Table 3 now lists experimental and predicted pK_a values, whilst Figure 4 summarises the statistics.

Nonetheless this is a very impressive body of work, that will be of considerable interest to workers in this area. In later work it would be of interest to have the authors expand further on just why this simple method works so well.

Reviewer #3 (Remarks to the Author):

In this paper, pKa is calculated for herbicide / therapeutic derivatives of 1,3-cyclohexanedione and 1,3-cyclopentanedione by utilising the linear relationship between bond length and pKa obtained from quantum chemical calculations. This is an interesting research and should be published in a general chemistry journal. However, there seem to be various opinions about whether or not to publish this paper in this journal, Comm. Chem.

One of the reasons is that the author have published a several papers by utilising the same relationship between the bond length and pKa (ref. 16-20). The target molecular species are diverse and differ in each paper, but they use essentially the same or similar computational procedure. The results obtained in the present study also seem to be of sufficient value for the molecules targeted for calculation. However, from the perspective of methodology novelty, it can be regarded as a routine work.

We have found this particular methodology to work very well on simple systems, and hence tested the hypothesis that the relationship is observed for more complex problems, like for tautomerizable compounds.

In addition, it should be pointed out that the relationship between bond length and pKa has been known for a long time. The following papers are examples;

A. J. Kirby, Adv. Phys. Org. Chem. 1994, 29, 87-183.

A. J. Green, J. Giordano, J. M. White, Aust. J. Chem. 2000, 53, 285-292.

J. E. Davies, N. L. Doltsinis, A. J. Kirby, C. D. Roussev, M. Sprik, J. Am. Chem. Soc. 2002, 124, 6594-6599

Moreover, the following might be in a similar position;

R. Fujiki et al. Phys. Chem. Chem. Phys., 2018, 20, 27272-27279.

and their related works.

All these papers have been added as references, thank you. Note that the last one does not involve bond lengths.

They are more or less related to linear free energy relationships. Indeed, the title of one of the authors' paper (Ref 19)

Linear Free-Energy Relationships between a Single Gas-Phase Ab Initio Equilibrium Bond Length and Experimental pK(a) Values in Aqueous Solution, indicating that this point is well recognized by the authors.

Roughly speaking, the direction of research progress based on computational chemistry can be divided into deductive-type based on physical chemistry and inductive-type on information/chemoinformatics. The importance of the latter seems to increase more and more from a pragmatic point of view. However, as described above, since this linear relationship is already well known, it is desired to systematically improve the prediction accuracy based on new ideas, for example, in information science.

It is our view that descriptors that encode relevant geometric features, that is, geometries that have a high probability of existing out of the available conformations, can be used to relate structure to property and potentially to activity. Highly probable geometric arrangements of atoms can only be determined by quantum mechanical calculations (which presumably falls into the physical chemistry category mentioned above). Hence, we don't really think that the importance of physical chemistry is lesser compared to what one can obtain from informatics (presumably 2D molecular fingerprints). We suggest that in future these two approaches might be combined, i.e. fingerprints that encode 3D information obtained via quantum mechanical calculations.

At the same time, one might feel that this linear relationship is obvious or self-evident.

Agreed, but under-utilised and not at all utilised in the context of the specific problem we address here.

But the reviewer does not agree this. In view of the difficulty of evaluating free energy in solution, it is not easy to explain this linear relationship sufficiently logically from a physicochemical standpoint. It is highly expected to approach from at least one of these two positions, physical chemistry or informatics.